# Morphology and Optical Properties of Gas-Phase-Synthesized Plasmonic Nanoparticles: Cu and Cu/MgO

**DOI:** 10.3390/ma15134429

**Published:** 2022-06-23

**Authors:** Sergio D’Addato, Matteo Lanza, Anthea Boiani, Eleonora Spurio, Samuele Pelatti, Guido Paolicelli, Paola Luches

**Affiliations:** 1Dipartimento di Scienze Fisiche, Informatiche e Matematiche, Università di Modena e Reggio Emilia, Via G. Campi 213/a, 41121 Modena, Italy; 243296@studenti.unimore.it (M.L.); 238229@studenti.unimore.it (A.B.); eleonora.spurio@unimore.it (E.S.); samuele.pelatti@unimore.it (S.P.); 2Istituto Nanoscienze-CNR, Via G. Campi 213/a, 41121 Modena, Italy; guido.paolicelli@nano.cnr.it (G.P.); paola.luches@unimore.it (P.L.); 3EN&TECH, Università di Modena e Reggio Emilia, Via G. Campi 213/a, 41121 Modena, Italy

**Keywords:** nanoparticles, plasmonic properties, copper

## Abstract

In this paper, an investigation of the properties of Cu and Cu/MgO nanoparticles (NPs) is presented. The NPs were obtained with gas-phase synthesis, and the MgO shells or matrices were formed via the co-deposition method on inert substrates. SEM and AFM were used to investigate the NP morphology on Si/SiO_x_, quartz, and HOPG. The Cu NPs revealed flattening of their shape, and when they were deposited on HOPG, diffusion and formation of small chains were observed. The embedding of Cu NPs in MgO was confirmed by TEM and EDX maps. XPS showed that Cu was in its metallic state, regardless of the presence of the surrounding MgO. UV–Vis revealed the presence of an intense localized surface plasmon resonance (LSPR) for Cu/MgO and for “bare” NPs. These results confirmed the role of MgO as a protective transparent medium for Cu, and the wavelength position of the LSPR in the Cu/MgO system was consistent with calculations. The wavelength position of the LSPR observed for “bare” and post-oxidized Cu NPs was probably affected by the formation of copper oxide shells after exposure to air. This study paves the way for the use of Cu/MgO NPs as plasmonic nanomaterials in applications such as photovoltaics and sensor technology.

## 1. Introduction

The study of plasmonic nanoparticles (NPs) and nanostructures is a very important subject of research in materials science for applications in electronics, optoelectronics [1], and sensor technologies, such as surface-enhanced spectroscopy [2], matrix-assisted laser desorption/ionization mass spectroscopy [3], and metamaterial-assisted biosensing. This last application is based either on strong coupling between surface plasmon polaritons and magnetic plasmon resonances or on magnetic resonances in refractive-index sensors [4,5]. Another very interesting subject is the use of plasmonic NPs in photovoltaic (PV) materials [6]: The insertion of plasmonic NPs during the fabrication of Si-based [7,8], thin films [9], and new generation solar cells have been proved to be very promising [10,11,12]. NPs can improve the light harvesting of solar cells using different geometries: when they are used as light scatterers on the front surface, in the cell active region, and at the interface between the charge transport area and the back contact. In particular, it has been proved that the power conversion efficiency of the device can increase [13,14]. Generally, the metals employed for synthesizing plasmonic NPs are Au and Ag [15]. While Au NPs are robust in environmental conditions, Ag ones can be contaminated because of oxidation in atmosphere, so they need to be shielded or encapsulated in transparent and inert shells [12,15]. On the other hand, the main advantage of Ag NPs is the reduced cost of the raw material. The routes for NP fabrication are generally based on chemical or physical synthesis procedures [16,17]. Chemical synthesis allows “scaling up” to industrial production, but physical synthesis can be an interesting alternative when “fine-tuning” of the NP properties is required. It can be a single-step and ligand-free process, allowing more accurate analysis of the NP structure, as well as the electronic, optical, and magnetic behavior. This analysis is of crucial importance for the design of new functional materials. Different physical synthesis procedures have been developed in recent years: top–down methods, such as e-beam lithography [18,19] and focused ion beam assisted deposition [20], and bottom–up methods, such as pulsed laser deposition [21] thermal evaporation with self-organization [22,23,24,25,26], and gas-phase synthesis [6,27,28,29]. The last method has been proved to be very flexible, because of the possibility of selecting the NP mass, generating nanoalloys [30], and developing core–shell geometries [31], which are of crucial importance if the protection of NPs from contamination is needed.

As previously established, Au and Ag NPs are the most studied systems in PV applications. On the other hand, Cu NPs can present some advantages, because of the low cost and versatility of the raw material. Indeed, researchers have turned their attention to gas-phase-synthesized Cu NPs for their interesting properties [32,33,34,35,36]. Cu NPs have been used for applications in photocatalysis, for instance, in H_2_–D_2_ exchange reaction, driven primarily by thermalized hot carriers [34], and to improve the performances of perovskite light-emitting diodes [35]. The main problem with the use of Cu NPs is their instability during air exposure, with consequent oxidation and loss of the localized surface plasmon resonance (LSPR). A possible solution is the generation of a stable oxide shell (mainly composed of Cu (II) oxide), with ozonization [36] or thermal treatment [21]. An alternative one-step procedure is the co-deposition of the pre-formed NPs with an inert and transparent material that can act either as a matrix or as a shell. This method has been successfully tested on Ag NPs co-deposited with MgO [37] and CaF_2_ [38]. Ag/MgO NPs produced in this way were also deposited on the porous TiO_2_ layer of perovskite solar cells, giving rise to an increase in PCE by 5% [15]. Although Cu NPs/MgO systems have been previously studied, MgO has mainly been used as an inert substrate [3,21,34,35] In this study, Cu NPs and Cu/MgO NPs were generated with a magnetron-assisted gas aggregation source (GAS), and they were co-deposited with MgO by making use of reactive thermal evaporation of Mg in O_2_. The chosen substrates were Si with its native oxide (Si/SiO_x_), highly oriented pyrolytic graphite (HOPG), and quartz. The morphology of the NP assemblies was investigated ex situ with scanning electron microscopy (SEM), atomic force microscopy (AFM), and transmission electron microscopy (TEM), obtaining information on NP size and shape, and for the creation of MgO shells, also checked with energy dispersion X-ray (EDX) maps. In situ X-ray photoelectron spectroscopy (XPS) was used to investigate the chemical state of “bare” and MgO-encapsulated Cu NPs. Finally, ultra-violet–visible (UV–Vis) optical spectroscopy gave evidence of the presence of a stable LSPR in Cu/MgO NPs, and its wavelength was compared with theoretical calculations.

## 2. Materials and Methods

The Cu NP samples were physically synthesized with a magnetron-assisted GAS NC200U (Oxford Applied Research), and the charged nanocluster beam was focused by means of the ion optics of a quadrupole mass filter (QMF) on a substrate in vacuum. The NP source and the QMF were attached to an ultra-high vacuum system comprising a deposition chamber (base pressure *p* = 9•10^−9^ mbar) equipped with a Mg thermal evaporator and a quartz microbalance, and an analysis chamber (*p* = 5•10^−10^ mbar) with a dual-anode X-ray source and a hemispherical electron energy analyzer (SPECS HA200), for in situ XPS analysis. Details of the experimental system can be found in [29]. The GAS discharge was kept fixed at V = 200 V, I = 0.21 A, with an Ar gas flow f = 60 sccm. The measured deposition rate *r* on the substrates varied between 0.1 and 0.4 nm/min, depending on the Cu target conditions. Here, *r* is defined as the thickness of a Cu solid film deposited on a unit area per unit time. For NP oxidation experiments, two procedures were mainly used: flowing oxygen gas in the aggregation region of the NP source with a flow *f* = 5 sccm [39] or flowing O_2_ in the deposition chamber (pressure in the chamber was *p* = 2•10^−7^ mbar for a time *t* = 20′) *after* the Cu NP film was formed on the substrate (post-oxi samples). Deposition or co-deposition of MgO on the sample was obtained by thermally evaporating Mg in presence of O_2_ (*p* = 2•10^−6^ mbar), with a Mg deposition rate varying between *r* = 0.4 and *r* = 0.6 nm/min. After deposition procedures, the samples were transferred in vacuum to the attached analysis chamber for XPS measurements. The substrates used during the experiments were Si/SiO_x_ wafers and freshly exfoliated HOPG crystals for SEM measurements, quartz slabs for AFM and optical experiments, and C-coated Ni grids for TEM observations. After the NP preparation and XPS experiments, the samples were transferred in in atmosphere to other analysis facilities. In particular, the lateral size distribution and the amounts of deposited NPs were measured with SEM on Si substrates, while the NP height distribution was checked with AFM. The NP lateral size distribution and morphology could not be measured when the quartz supports were used, because of the insulating nature of this substrate, which hampered the SEM analysis. The SEM experiments were performed using a Nova Nano SEM450 (FEI Company-Bruker Corporation, Billerica, MA, USA). The SEM column was equipped with a Schottky field-emission gun (SFEG), and it could achieve a resolution of 1.4 nm in a low-voltage (1 KV) operation. AFM images were taken with an NTEGRA AURA model, NT-MDT. The TEM measurements were carried out with a TALOS F200S G2 (Thermo-Fisher Scientific, Waltham, MA, USA) equipped with a Schottky field emitter (80–200 keV) operating in TEM and STEM modes, and a double silicon drift detector (SDD) for energy dispersive X-ray (EDX) spectroscopy and mapping. Finally, UV–Vis optical spectroscopy was carried out with a spectrophotometer equipped with a Xe lamp, a grating monochromator, a linear polarizer, and a silicon photodetector (detection range 250–750 nm). The absorbance A was evaluated by measuring the transmittance T and the reflectance R as A = 1 − T − R. The calculations of the optical constants were performed with the commercial Igor Pro software package.

## 3. Results and Discussion

### 3.1. NP Morphology

In this subsection, the findings of SEM, TEM, and AFM experiments on Cu and Cu/MgO NPs are revealed, and their morphology is discussed. Figure 1a shows an SEM image of Cu NPs deposited on Si/SiO_x_, while Figure 1b illustrates the associated size distribution fitted with a lognormal curve. At the low coverage fraction *C* of Figure 1a (*C* = 4%, corresponding to a nominal thickness *t* = 0.5 nm, as measured by the thickness monitor), the NPs were well-separated, and they showed mostly circular shapes. The formation of a few dimers and trimers is shown by red closed lines in Figure 1a. The fitting parameters yielded an average lateral size <*d*> = 11.4 nm, with a full-width half-maximum (FWHM) value Δ*d* = 3.3 nm. With increasing coverage, the number of agglomerates increased, as can be observed in Figure 1c, (*t* = 2.4 nm, *C* = 15%), and the NPs tended to coalesce, in a similar fashion to what was observed previously on Ag NPs [37].

This tendency is confirmed by the TEM images shown in Figure 2. In Figure 2a, a bright-field image of low-coverage Cu NP films is shown, while Figure 2b clearly reveals that the Cu NP density significantly increased. The formation of agglomerates in the second case is evident, and they can also be ascribed to higher mobility of the NPs on the C film of the TEM sample support when compared with Si/SiO_x_ substrates used in the SEM analysis.

The diffusion and agglomeration of NPs on carbonaceous supports (especially HOPG) were previously studied [40,41]. This phenomenon was also investigated in the present study. Figure 3 illustrates an SEM image of Cu NPs deposited on a freshly exfoliated HOPG crystal. The Cu NPs decorated the steps, in a similar way to what occurs to adatoms on crystalline surfaces, and they also formed elongated structures, resulting from diffusion after landing. In principle, the experimental measurement of the diffusion coefficient *D* can be performed on different systems, such as molecules in porous systems with thermogravimetry [42] or adatoms during the growth of thin films using time-resolved X-ray reflectivity and scattering [43]. The diffusion of atoms on the surface has also been extensively studied with scanning tunneling microscopy [44,45]. In the case of deposition of *pre-formed* NPs, a direct measurement of *D* for NPs on surfaces was not possible, although the mobility of large clusters (up to tens of thousands of atoms) was experimentally demonstrated with SEM and TEM. In a previous study [37], a sequence of TEM images was acquired during the movement of two Ag NPs that eventually coalesced. By using a procedure similar to the one described in [40], it was possible to obtain an estimate for the diffusion coefficient D from the following Equation (1) [40]:(1)$$D={\left(\frac{0.41}{N_{agg}}\right)}^{\frac{1}{\chi }\;}F\pi d^{4}/16$$
where *N_agg_* is the average number of NP aggregates (or “islands”), as obtained by counting the number of agglomerates composed of two or more NPs and normalizing them to the number of possible sites of nucleation in the same image [40], assumed to be the maximum number of particles of diameter *d* that can be arranged in close-packed surface geometry. *F* is the number of incident NPs on a unit surface area, *d* is the average particle diameter, and *χ* = 0.336. For *N_aggr_* ≈ 3%, *d* = 11.4 nm, and *F =* 0.1 NP/s cm^2^, a value *D ≈* 8•10^−11^ cm^2^ s^−1^ was obtained. This quantity is of the same order of magnitude as the one obtained for Ni NPs [41] but smaller (by a factor of 12) than in the case of Sb NPs with *d* = 6 nm on carbon films, found by Bardotti et al. [40], where *D* ≈ 10^−9^ cm^2^/s at room temperature. The reason for this discrepancy can be the use of different substrates for the two experiments, a higher degree of adhesion of Cu NPs to the surface than Sb, and the much larger size of Cu NPs in the present experiment. For instance, a larger size implies a much larger mass (M = 2.8•10^6^ AMU for Cu NPs in the present experiment), compared with M = 4.5·10^5^ AMU for Sb in the experiment cited in [40], hence the reduced mobility.

The effect of co-deposition of Cu NPs and Mg in an oxygen atmosphere on the NP film morphology is clearly shown in Figure 4, where a SEM image obtained from Cu nanoclusters and MgO co-deposited on Si/SiO_x_ is shown. The formation of islands with approximately a square or rectangular shape is evident, with a lateral size ranging between 20 and 50 nm, a much larger value than the average value of *d* for the original Cu NPs. These shapes are clearly an effect of the formation of MgO shells/matrices embedding the Cu NPs, in a similar way to what was observed on the Ag/MgO [37] system, obtained by using the same apparatus and synthesis method. On the basis of previous results [37,38], as well as the XPS data found in this study (see the next subsection), it can be inferred that this peculiar arrangement is essentially due to two main reasons: (1) a higher reactivity of Mg to O species, resulting in the preferential oxidation of Mg compared with the metal clusters during co-deposition, and (2) a higher sticking coefficient of MgO to metal NPs than inert substrates, such as Si/SiO_x_ and carbon films used in TEM experiments. This synthesis recipe results in either the formation of metal-core/MgO-shell NP structures or in metal NPs embedded in a MgO matrix, depending on the NPs and Mg atoms flux during co-deposition.

Bright-field TEM and EDX mapping experiments were performed on Cu/MgO NP samples. In the image presented in Figure 5a, the Cu clusters (darker areas) already exhibited some agglomeration, with MgO embedding them (clearer areas). EDX maps allowed a clearer distinction between the different species, as observed in Figure 5b–d, where Cu, Mg, and Cu/Mg maps are shown in false colors: Mg signal was rather diffused, but its intensity was higher in the regions around Cu NPs, confirming the formation of a core–shell or core–matrix structure.

Another important aspect concerns the vertical height of the deposited Cu NPs, as the optical spectra are strongly influenced by the shape of the NPs when they are deposited on a support [26,46]. Figure 6 shows an AFM image of Cu NPs deposited on a quartz substrate; the choice of this substrate was made to allow for optical measurements in transmission geometry. The vertical height distribution obtained via a detailed grain analysis of the AFM in Figure 6a is plotted in Figure 6b. A fitting of the main peak of the distribution with a Gaussian profile gave an average height <*h*> = 7.5 nm, with an FWHM value equal to Δ*h* = 2.2 nm.

By comparing this value to the average lateral diameter obtained from the SEM image of Figure 1a, an aspect ratio (AR) = <*d*>/<*h*> = 1.5 was obtained. The “flattening” of the bare metal NPs deposited on substrates was observed before [47,48,49]. The reason for this effect can be ascribed to the agglomeration and coalescence of NPs and to their interaction with the support [47,48,49]. This deformation affects the optical properties, as described in Section 3.2.

### 3.2. NP Electron and Optical Properties

XPS analysis was carried out on Cu and Cu/MgO NPs deposited on quartz substrate before the optical experiment. In particular, the Cu 2p and Cu L_3_VV line shapes were taken, to obtain information about the chemical state of Cu atoms in the NP. Due to the poor electrical conductivity of the quartz substrate, the samples were electrically charged during the XPS experiment, and the binding energy (BE) positions were realigned by calibrating them to the Si 2p signal in SiO_2_ (BE = 103.3 eV). Figure 7 illustrates Cu 2p and Cu L_3_VV spectra taken with the Al K_α_ emission line (photon energy *h**ν* = 1486.7 eV) from three different samples: as-deposited Cu NPs, Cu NPs co-deposited with MgO, and Cu NPs after exposure to oxygen in the deposition chamber (*p* = 2•10^−7^ mbar, time of exposure Δ*t* = 30′). The as-deposited Cu NPs spectra showed the typical line shape of metallic Cu [32], and the same aspect can be observed for the Cu/MgO sample spectra. At variance, the line shapes of the photoemission and Auger spectra after exposure to oxygen presented strong changes. In this sample, the appearance of a second doublet in the 2p signal, as well as a broadening of the L_3_VV, are evident, with the loss of the fine structures accompanying the typical emission line, characteristic of the metallic Cu spectrum [32]. From these results, it can be deduced that Cu NPs retained their metallic state if oxygen was co-deposited with Mg, resulting in the formation of a MgO shell or matrix surrounding the Cu core.

The optical properties of bare Cu and Cu/MgO NPs were also investigated with UV–Vis spectroscopy. UV–Vis absorbance spectra taken at p-polarization with an incidence angle of Θ = 22° are plotted in Figure 8. The four spectra were, respectively, taken from bare Cu, Cu/MgO, and Cu after exposure to oxygen (labeled as post-ox) in the deposition chamber, and oxidized Cu NPs obtained by flowing oxygen in the aggregation region (*f* = 5 sccm) of the nanocluster source. The corresponding XPS spectra of the first three samples are shown in Figure 7. The presence of a distinct resonance in the wavelength region between λ = 570 nm and λ = 590 nm is evident for Cu/MgO, Cu, and Cu post-ox, and it is indicated with a red arrow, while such structure is missing in the absorbance spectrum taken from Cu NPs oxidized in the aggregation region. On the basis of previous experimental results [21,36], this feature can be assigned to the LSPR of the Cu NPs. It is interesting to note that this structure was more intense in the Cu/MgO sample and blue-shifted compared with bare Cu and post-ox Cu. Its absence in the absorbance of the Cu NPs exposed to O in the aggregation region can be seen as experimental evidence of their complete oxidization.

For a better understanding of the observed optical properties, the imaginary part of the optical polarizability α was calculated for bare Cu and Cu NPs embedded in MgO by assuming a spherical and an oblate ellipsoidal geometry, with different values of the ratio between the major and minor axes, that should be identified with the experimentally found AR. The calculations were carried out by using the Maxwell–Garnett model [46,50]. The Maxwell–Garnett equation of the polarizability of an ellipsoid is
(2)$$\alpha_{i}=\frac{4}{3}\pi abc\varepsilon_{0}\frac{\varepsilon -\epsilon_{m}}{\varepsilon_{m}+\left[\varepsilon -\varepsilon_{m}\right]\cdot L_{i}}\;$$
where *a*, *b*, and *c* are the ellipsoid semiaxes; *ε*_0_ is the dielectric constant in the vacuum; *ε* is the complex dielectric function of the NP material; *ε_m_* is the dielectric function of the medium surrounding the NP; *L_i_* is a depolarization factor; and *i* = *x*, *y*, *z.* While *x* and *y* are the directions parallel to the axes *a* and *b* of the ellipsoid (and parallel to the surface of the support), *z* is the direction normal to the surface. In the case of a sphere, the model yielded *L_i_* = 1/3 for the three components, *a = b = c = r* (corresponding to the sphere radius), *α_x_* = *α_y_* = *α_z_* = *α*. Equation (2) has the following form:(3)$$\alpha =12\pi r^{3}\varepsilon_{0}\frac{\varepsilon -\varepsilon_{m}}{2\varepsilon_{m}+\varepsilon }\;$$

In the case where *a* = *b* > *c*, we have an oblate ellipsoid, representing a model for the NPs with an aspect ratio AR = *a*/*c* = *b*/*c* > 1. It can be assumed that the NPs have this shape, according to the findings of the SEM and AFM measurements. For AR = 1.5 we have *L_x_* = *L_y_* = 0.272, *L_z_* = 0.455, so the components of the polarizability are different, *α_x_* = *α_y_* ≠ *α_z_*. Additionally, when AR=2, the depolarization factors is *L_x_ = L_y_ =* 0.236, *L_z_ =* 0.527. These differences in the components of *α_i_* are the cause of the shifts in the position of the LSPR for different orientations of the incident polarization vector of the radiation [26,46,50].

In the present calculations, it is assumed that Cu NPs are embedded in MgO, as suggested by the SEM and TEM images, with the Cu and MgO dielectric functions obtained in the literature [51,52]. The results for this model representing the optical properties of Cu/MgO NPs are shown in Figure 9, and the experimentally found positions of the LSPR and the theoretical results are reported in Table 1 and Table 2. The “spherical” model (i.e., AR = 1) showed a resonance, with a maximum at λ = 573 nm, with a tail at a short wavelength, caused by the contribution of interband transitions to the Cu dielectric function.

As regards the “oblate ellipsoidal model”, the maxima for AR = 1.5 were positioned at λ = 584 nm and λ = 564 nm for *Im*(*α_x_*) and *Im*(*α_z_*). Regarding the simulations for AR = 2, the shape of the obtained curves is very similar, and the maxima fell at λ = 601 nm for Im(α_x_) and λ = 554 nm for Im(α_z_). Comparing these theoretical results with the obtained optical spectra, it can be deduced that there was a reasonable agreement in the position of the LSPR, with a value of λ lying between the observed maxima for the model with AR = 1.5. It should be borne in mind that, in the adopted experimental geometry (p-polarization with an incidence angle of Θ = 26°), the spectrum resulted from a combination of the two components of the polarizability. It can be deduced that the combination of the two components of the imaginary part of polarizability for AR = 1.5 would give a position for the LSPR at intermediate values between λ = 584 nm and λ = 564 nm, which is the wavelength range under which the experimental position of the LSPR fell (λ_exp_ = 572 nm), so an oblate Cu ellipsoid with AR = 1.5 and embedded in MgO is a reasonably accurate model to describe the optical properties of this system.

In the case of bare and post-ox NPs, the position of the LSPR in the experimental data was slightly red-shifted to λ = 586 nm and λ = 592 nm, respectively. The calculated Im(α) for a sphere in vacuum resulted in a resonance positioned at λ = 551 nm, i.e., a different value from that observed in the experimental one. When the AR changed to AR = 1.5, the positions of the LSPR fell at λ = 563 and λ = 545 nm, respectively. Therefore, there was a moderate discrepancy between the experimental position of the LSPR and the theoretical one for bare Cu NPs. On the other hand, the LSPR feature in the data taken from post-ox Cu NPs fell at a similar wavelength as the one in the bare Cu NPs. Since the optical measurements were performed in the air after a few hours, this behavior can be probably explained by partial oxidation of the bare Cu NPs when exposed to air, and the optical data were found to be similar to the data of the NPs that were *intentionally* oxidized. The formation of an oxide shell around the metallic Cu NP core probably gave rise to the redshifted LSPR compared with the (theoretical) Cu and Cu/MgO systems. It must be stressed that in situ XPS analysis showed that bare Cu samples were in a clean metallic state after deposition (see Figure 7).

Overall, the microscopy results obtained on the Cu/MgO NP samples, the XPS data, and the intensity and position of the LSPR in the absorbance spectrum strongly indicate that MgO constitutes a good protective and transparent material for the Cu NPs, which are otherwise subject to oxidation under exposure to air. This result was also found for Ag/MgO [37] and Ag/CaF_2_ [38] NPs that were grown using the same method. The formation of shells with a predominance of Cu(II) oxide around metallic Cu core was also found to be an effective way to block the effect of progressive oxidation [21,36]. However, the formation of such a protective shell requires thermal treatment and/or use of ozone in UV irradiation, while the co-deposition method is less elaborate, and the size of the original NP remains the same, while the Cu(II) oxide shell is formed at the expense of the Cu core size. The NPs obtained in this way can have interesting applications in plasmonics, photovoltaic materials, and photocatalysis.

## 4. Conclusions

In this study, an experimental investigation of morphological and optical properties was carried out on gas-phase-synthesized Cu and core–shell Cu/MgO NPs. The SEM, AFM and TEM data showed Cu NPs with an average lateral size <*d*> = 11.4 nm (size dispersion Δ*d* = 3.3 nm) and an average height <*h*> = 7.5 nm, with a dispersion Δ*h* = 2.2 nm. An aspect ratio AR = 1.5 was found, caused by the deformation of the NPs due to their interaction with the substrate and by the presence of small agglomerates formed during deposition. By increasing coverage, the formation of agglomerates became more evident. When Cu NPs were deposited on freshly exfoliated HOPG, diffusion and aggregation with the formation of islands occurred, and this was also evident in the preferential adsorption on surface steps. A diffusion coefficient *D* = 8•10^−11^ cm^2^/s was obtained. In the case of co-deposition of Cu NPs with MgO on Si/SiO_x_, SEM images showed approximately square or rectangular islands, due to the growth of MgO around the metal NPs. TEM images and EDX maps confirmed that MgO formed a matrix embedding the Cu NPs. At the same time, in situ XPS results of Cu 2p and Auger LVV demonstrated that Cu was still in its metallic state. Finally, UV–Vis optical absorbance spectra showed the presence of a distinct feature that was ascribed to the Cu LSPR, which was confirmed by calculations of the polarizability using the Maxwell–Garnett model for an oblate ellipsoid embedded in MgO. These results demonstrate the efficacy of the co-deposition method of Cu NPs with MgO in preventing the contamination of the metallic NP, preserving the presence of the LSPR, and opening a pathway to the use of this nanomaterial in devices such as sensors and PV cells.

## Figures and Tables

**Figure 1 materials-15-04429-f001:**
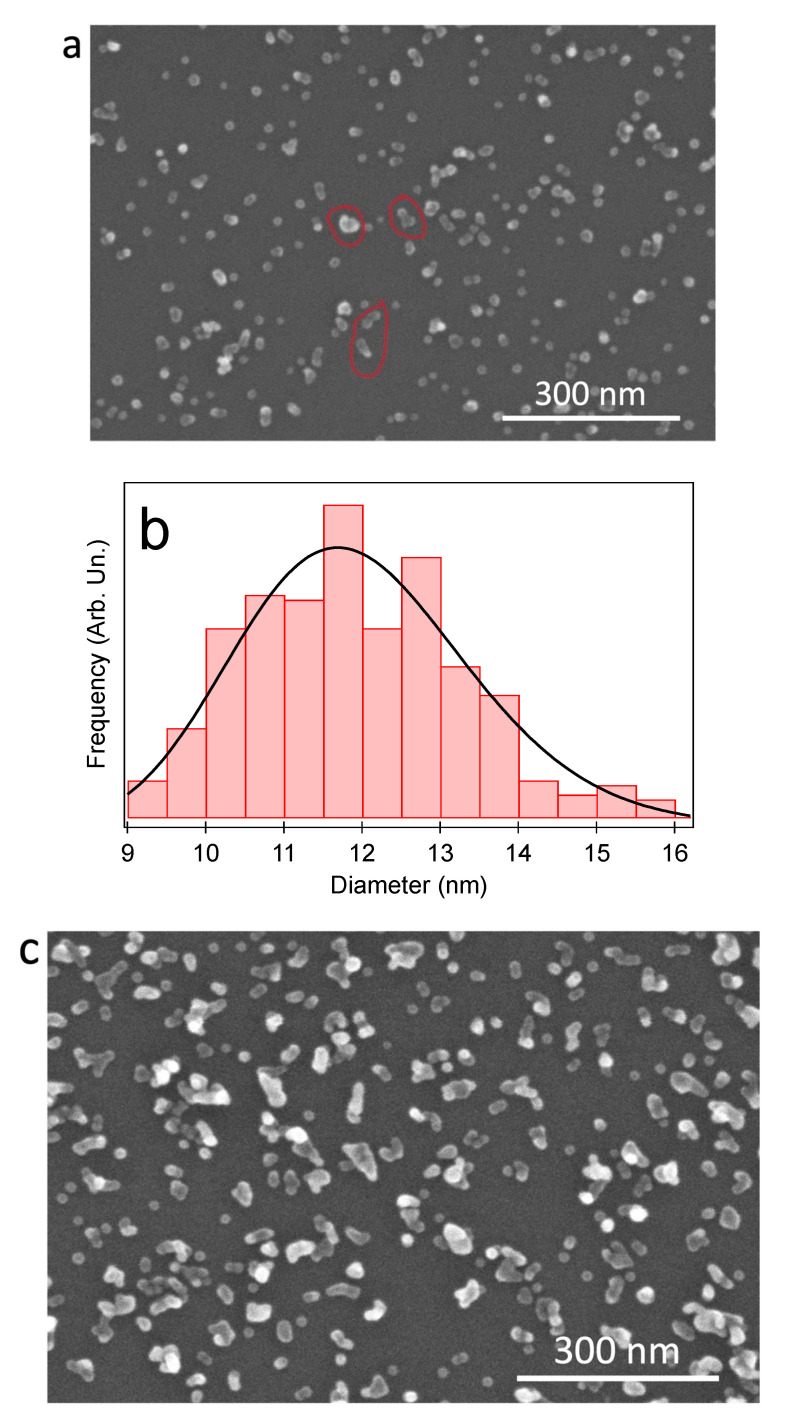
(**a**) SEM image of Cu NPs deposited on Si/SiO_x_ with nominal thickness *t* = 0.5 nm, fractional coverage *C* = 4%; (**b**) lateral size distribution of the NP assembly shown in (**a**); (**c**) SEM image of Cu NPs deposited with *t* = 2.4 nm, *C* = 15%.

**Figure 2 materials-15-04429-f002:**
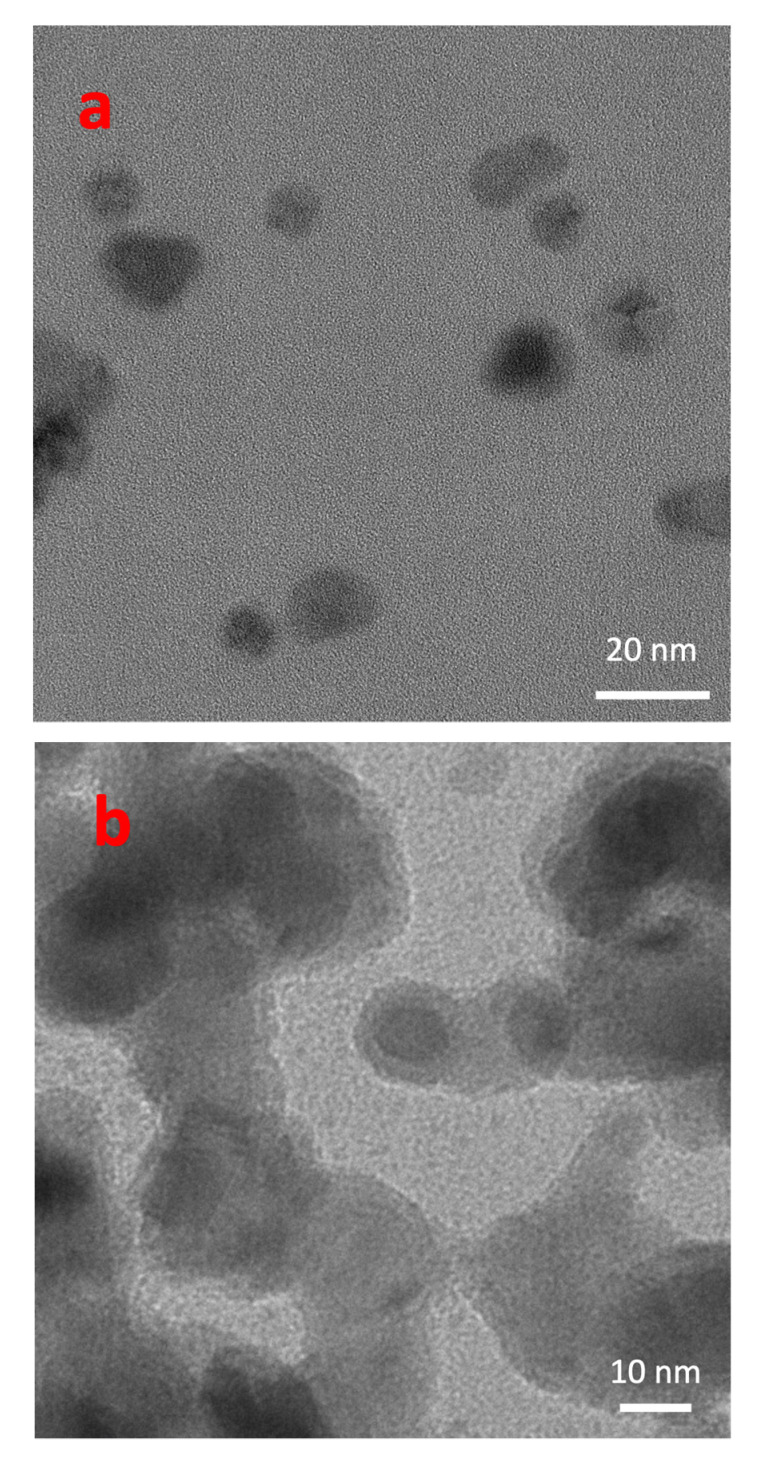
Bright-field TEM images of bare Cu NPs at (**a**) low coverage (*C* = 2%, *t* = 0.2 nm) and (**b**) high coverage (*t* = 6.7 nm). In this case, the fractional coverage is not a meaningful parameter, as the NPs agglomerated also along the direction normal to the surface.

**Figure 3 materials-15-04429-f003:**
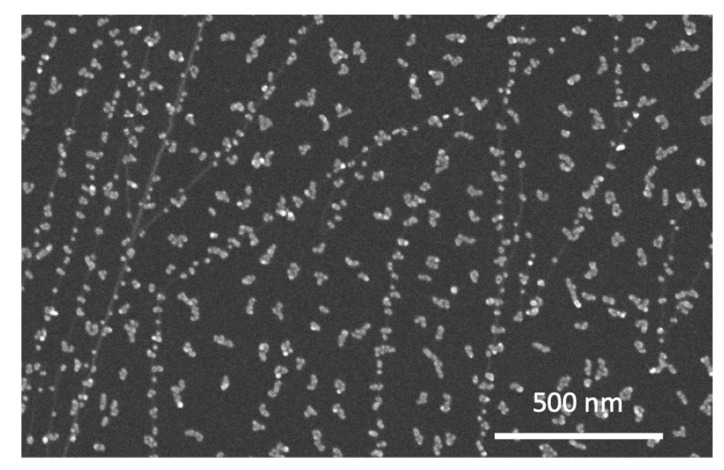
SEM image of Cu NP deposited on HOPG. *t* = 0.2 nm, *C* = 4%.

**Figure 4 materials-15-04429-f004:**
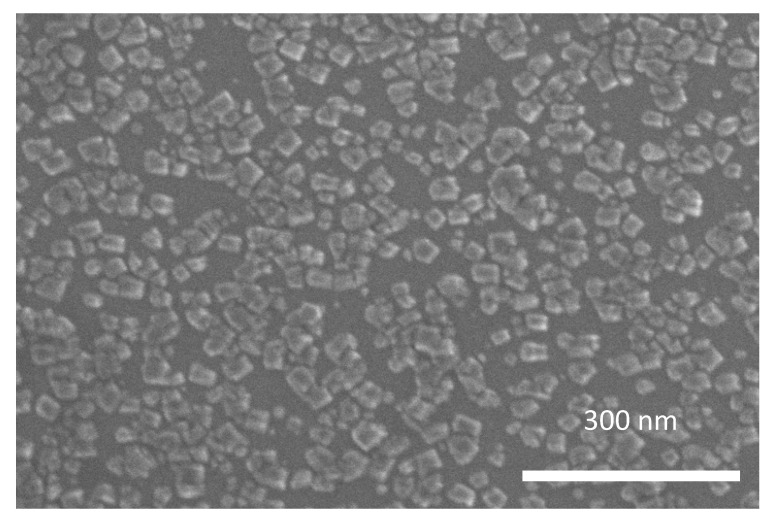
SEM image of Cu NP co-deposited with Mg in O atmosphere. The nominal thickness for Cu NP and MgO are *t*_Cu_ = 0.4 nm, *t*_MgO_ = 1 nm.

**Figure 5 materials-15-04429-f005:**
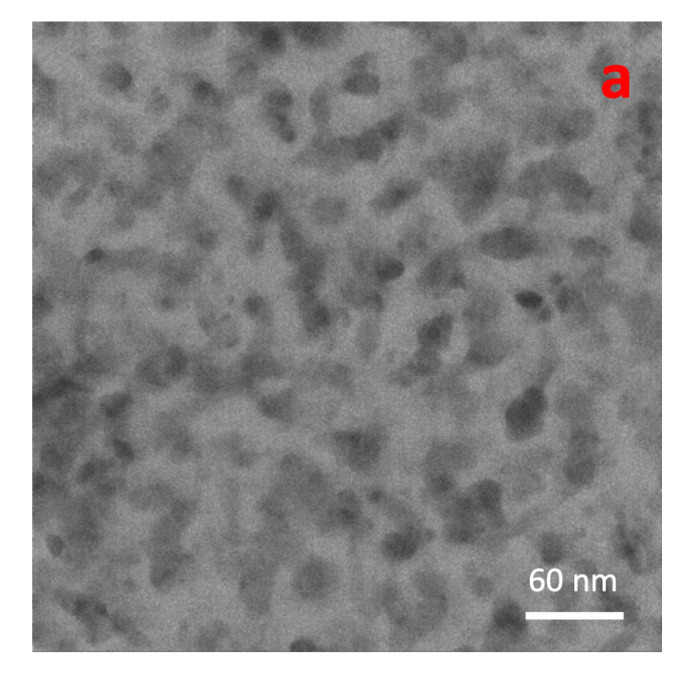
(**a**) Bright-field TEM image of Cu/MgO NP film obtained with co-deposition method; (**b**) EDX map obtained by measuring the intensity of the Mg EDX signal; (**c**) EDX map of Cu signal; (**d**) combination of maps (**b**) and (**c**).

**Figure 6 materials-15-04429-f006:**
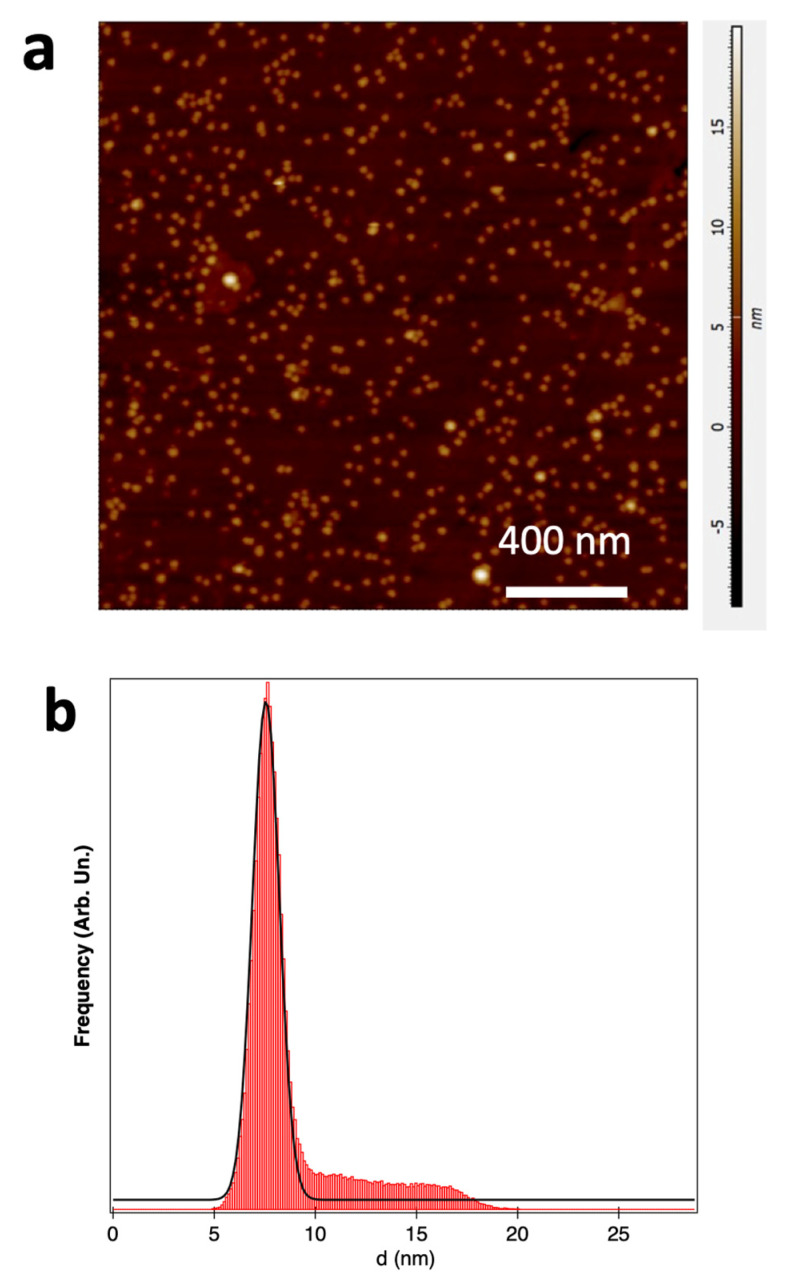
(**a**) AFM topography image of Cu NPs deposited on quartz, with nominal thickness *t* = 0.36 nm. The fractional coverage *C* was not measured, because of the limited lateral resolution of the AFM technique; (**b**) vertical height distribution of Cu NPs obtained via a grain analysis of image a (red histogram) and fitting curve (black line).

**Figure 7 materials-15-04429-f007:**
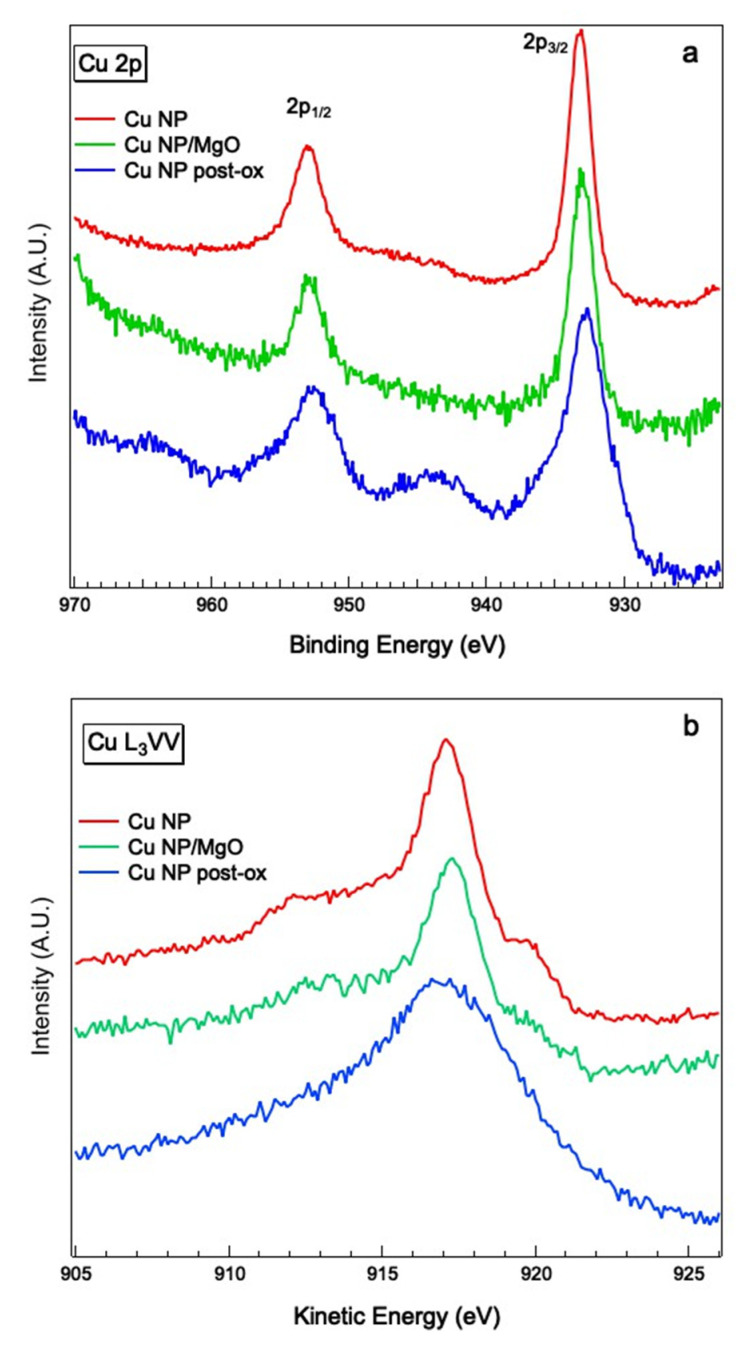
(**a**) Cu 2p XPS spectra obtained from Cu NPs (red line), Cu/MgO NPs (green line), and Cu NPs after exposure to oxygen in the deposition chamber (blue line); (**b**) Auger Cu L_3_VV from the same samples shown in (**a**).

**Figure 8 materials-15-04429-f008:**
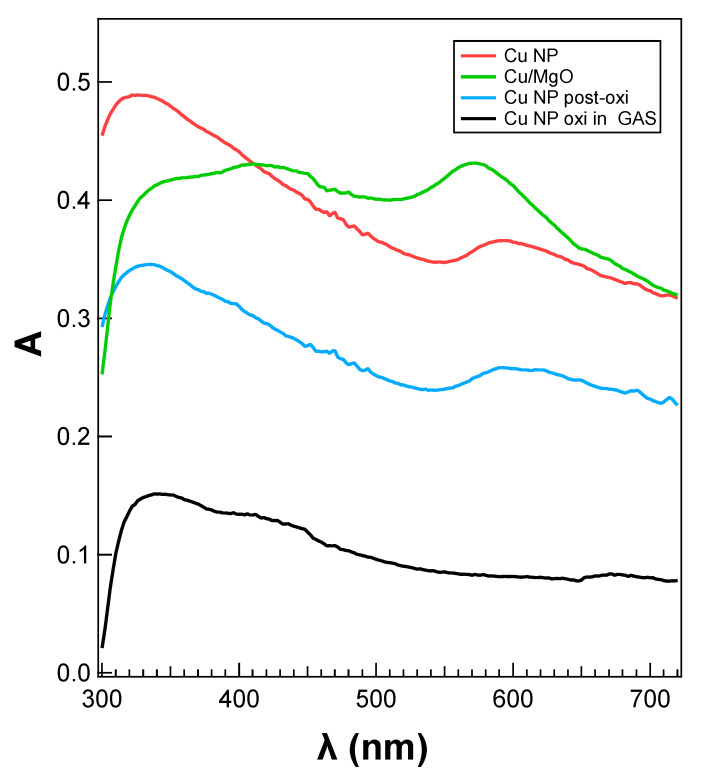
UV–Vis absorbance spectra from Cu/MgO NPs, bare Cu NPs, Cu NPs post-oxidized, and Cu NPs completely oxidized by flowing O_2_ in the GAS. The position of the SPR (indicated with red arrows) was blue-shifted for Cu/MgO compared with the bare and post-oxidized Cu.

**Figure 9 materials-15-04429-f009:**
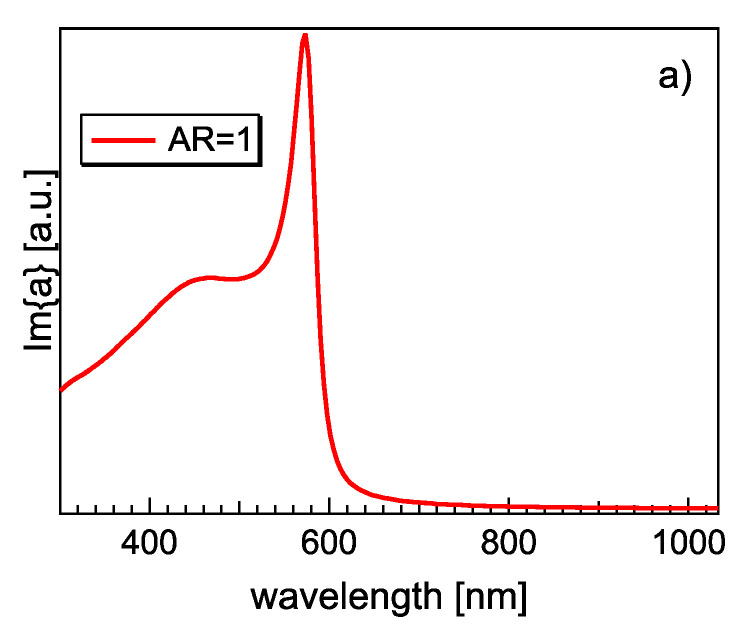
(**a**) Calculated imaginary part of the optical polarizability *Im*(*α*) from the Maxwell–Garnett model of a spherical Cu NP embedded in MgO (AR = 1); (**b**) *x* component *Im*(*α_x_*) calculated for an oblate ellipsoid with AR = 1.5 and AR = 2. *x* is the direction along the major axis, parallel to the surface of the support; (**c**) *z* component *Im*(*α_z_*), normal to the surface for AR = 1.5 and AR = 2.

**Table 1 materials-15-04429-t001:** Experimental and theoretical wavelength positions of the LSPR for Cu/MgO NPs. The results for AR = 1.5 and AR = 2 are, respectively, maxima of *Im*(*α_x_*) and *Im*(*α_z_*).

	Cu/MgO (exp)	Cu/MgO Im(a)	Cu/MgO Im(α_x_)	Cu/MgO Im(α_z_)	Cu/MgO Im(α_x_)	Cu/MgO Im(α_z_)
AR		1	1.5	1.5	2	2
l_p_ (nm)	572	557	583	557	601	554

**Table 2 materials-15-04429-t002:** Experimental wavelength positions of the LSPR for Cu and Cu post-ox NPs, and theoretical positions of the LSPR calculated for Cu NPs in vacuum. The results for AR = 1.5 are, respectively, maxima of *Im*(*α_x_*) and *Im*(*α_z_*).

	Cu (exp)	Cu (post-oxi, exp)	Cu (th)	Cu Im(α_x_)	Cu Im(α_z_)
AR			1	1.5	1.5
l_p_ (nm)	586	592	551	563	545

## Data Availability

Data supporting the reported results are available by request from S.D.

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
