# Peer review of "Morphology and Optical Properties of Gas-Phase-Synthesized Plasmonic Nanoparticles: Cu and Cu/MgO"

_materials, 2022, doi:10.3390/ma15134429_

Round 1

Reviewer 1 Report

In this work, the authors presented an experimental investigation of morphological and optical properties on gas-phase synthesized Cu and core-shell Cu/MgO nanoparticles. Ultra-Violet-Visible (UV-Vis) optical spectroscopy gave evidence of the presence of a stable localized surface plasmon resonance in Cu/MgO nanoparticles, and its wavelength was compared with theoretical calculations. This topic is interesting and the work opens the way to the employment of Cu/MgO nanoparticles in application of Cu plasmonic nanomaterials. The main results are clearly presented. In my opinion, this paper deserves publication in Materials, if the authors reasonably address the following comments:

1. The level of English should be improved, as many sentences are badly constructed.

2. There have been some relevant reports on the excitation of localized surface plasmon resonance in metallic nanostuctures and their related applications, especially for biosensing, such as Applied Physics Express 12(5), 052015 (2019); Results in Physics 14 (2019) 102397. They should be properly cited in the introduction part to give audience a broader picture of this field and improve the quality of your paper.

3. The authors used a procedure to obtain an estimate for diffusion coefficient D. The cauculted value of D has a large discrepancy as compared with previous experimental results. I suggest the authors to expeimetally measure diffusion coefficient D.

4. In Table 1 and Table 2, the authors compared the wavelength positions of LSPR from experments and theoretical calculations. However, the difference is obvious, which should be further explained.

Author Response

Reply to referee 1

We thank the referee for her/his comments and suggestions. Here is our reply. The revised part of the manuscript is written in red.

  1. We believe that we improved the level of English, especially In the Introduction and in the data discussion section.
  2. We included the citations suggested by the referee and other references, to give a broader view of the applications of plasmonic materials.
  3. There is indeed a discrepancy between the diffusivity obtained from our data and the ones obtained by Bardotti et al. (ref. 40). We tried to give some reasonable explanations of this: the different NP materials, the different mass value and the different substrate used in the two experiments. This part of the manuscript has been extended to detail our explanation. A direct measurement of D for the systems under study (preformed NPs deposited on a surface) is quite difficult, and it cannot be performed in our laboratories.
  4. We gave a detailed explanation of the data reported in table I and table II. In particular, we believe that the position of the LSPR found experimentally in the Cu/MgO NP system is consistent with the values found using the Maxwell-Garnett model assuming a NP AR=1.5, as it falls between the two positions found for the two components of the calculated Imaginary part of the polarizability, Im(ax) and Im(az). Since our spectra were taken at an oblique incidence angle in p-polarization, they represent a mixture of the two components, corresponding to the two components of the polarizability. This discussion is now more detailed than in the previous version of our manuscript.

Reviewer 2 Report

Cu and Cu/MgO nanoparticles have been synthesized in many previously reported works, the novelty of this work must be described. Also, as mentioned in the introduction, it is suggested at least one of the applications of this nanoparticles must be presented.  

Author Response

Reply to Referee2

We thank the referee for her/his comments and suggestions. Here is our reply. The revised part of the manuscript is written in red.

We showed some examples of previous works on Cu and Cu/MgO. To our knowledge, MgO has always been used as a support, while with our method it forms a matrix surrounding the particles. This point is now explicitly addressed in our manuscript. We also mentioned few examples of application of plasmonic Cu NPs in optoelectronics, MALDI and photocatalysis with proper references.

Reviewer 3 Report

In view of the article titled: Morphology and optical properties of gas phase synthesized plasmonic nanoparticles: Cu and Cu/MgO, I would like to add some minor points needed to modify before publication. The article is lacking in novelty in application point of view. Few questions are mentioned as follow;

1.      Authors claim the successful fabrication of Cu and Cu/MgO nano particles via gas-phase synthesis method, but never showed the confirmation via XRD analysis or some other tool.

2.      In order to investigate morphology and optical properties, why did the authors only focus on Cu and Cu/MgO?  Some explanation or discussion should be added in introduction to make it strong candidate as compared to other highly efficient Au of Ag based nano-particles.

3.      Can the authors provide the source of the data presented in Figure 8?

4.       The authors need to discuss the Maxwell-Garnet model code used for calculations. 

5.       For the better understanding of observed optical properties, the authors must present the Maxwell-Garnet model equations used for the calculations about imaginary part of the optical susceptibility for bare Cu and Cu NPs embedded in MgO.

6.      Figure 9 shows the calculated imaginary part of optical susceptibility. A discussion should be added to compare the theoretical simulation results with obtained optical spectra.

Author Response

Reply to referee 3

We thank the referee for her/his comments and suggestions. Here is our reply. The revised part of the manuscript is written in red.

  1. The Cu and Cu/MgO NPs are directly imaged with spatially resolved microscopy techniques, like SEM, AFM and TEM, (see figures 1, 2, 3, 4, 5, 6 of the manuscript) and the images have given enough evidence of the formation of the NP and of their morphology, as described in detail in the subsection 3.1. We do not believe that XRD was necessary to clarify this point.
  2. We discussed in the introduction section some of the optical works performed on Au and Ag NPs, and we compared them explicitly with the optical properties of Cu NPs. Cu raw material has a lower cost, and it can be efficiently used as plasmonic material in many applications (see for instance ref.s 3, 21, 34,35 in the new version of the manuscript.
  3. The source of the data in figure 8 are present in the experimental section, where the optical experiment that we performed in our lab is described.
  4. We introduced explicitly the Maxwell-Garnett model in the new version of our manuscript, with the explicit formulae that we used to calculate the components of the imaginary part of the polarizability. The code is a simple string that can be inserted in the Igor Pro software package. The details of the M-G model can be found in ref.s 46 and 50 of the new version of our manuscript.
  5. See point 4 in this reply.
  6. We gave a detailed explanation of the data reported in table I and table II. In particular, we believe that the position of the LSPR found experimentally in the Cu/MgO NP system is consistent with the values found using the Maxwell-Garnett model assuming a NP AR=1.5, as it falls between the two positions found for the two components of the calculated Imaginary part of the polarizability, Im(ax) and Im(az). Since our spectra were taken at an oblique incidence angle in p-polarization, they represent a mixture of the two components, corresponding to the two components of the polarizability. This discussion is now more detailed than in the previous version of our manuscript.

Round 2

Reviewer 2 Report

the manuscript is well revised.